# Integrating Diverse Cover Crops for Fallow Replacement in a Subtropical Dryland: Implications on Subsequent Cash Crop Yield, Grain Quality, and Gross Margins

**Ismail Ibrahim Garba** [1,2,*] and **Alwyn Williams** [1]

1    School of Agriculture and Food Sciences, The University of Queensland, Gatton, QLD 4343, Australia
2    Centre for Dryland Agriculture, Bayero University Kano, Kano 70001, Nigeria
*    Correspondence: i.garba@uq.net.au; Tel.: +61-0478-793-244

**Abstract:** Integrating cover cropping into crop–fallow rotation has been considered a key component of ecological intensification that could mitigate negative productivity and sustainability challenges associated with conventional fallow practices. However, the adoption of cover crops in water-limited environments has been limited by potential soil water and nitrogen (N) costs and resulting yield penalties. We examined the impacts of diverse cover crops on fallow soil water and mineral N dynamics and the legacy impacts on subsequent cash crop productivity and profitability. The cover crops used (forage oat—*Avena sativa* L. [grass], common vetch—*Vicia sativa* subsp. *sativa* L.)/fababean—*Vicia faba* L. [legume], forage rape—*Brassica napus* L. [brassica]) differed in functional traits related to growth, phenology, and soil water and N acquisition and use strategies. We found that grass-associated cover crops generally supported higher cash crop grain yield and profit than brassica- or legume-associated cover crops, mainly due to moderate biomass accumulation and water use and persistent groundcover. Cash crop grain yields increased by +19% and +23% following forage oat cover crop, with concomitant gains in gross margins of +96\$ ha$^{-1}$ and +318\$ ha$^{-1}$ for maize and winter wheat compared to conventional fallow. In contrast, maize grain yield following brassica-associated cover crops ranged from +8 to −21% and reduced gross margins by −229 to −686\$ ha$^{-1}$ relative to conventional fallow. Legume- and brassica-associated cover crops had the lowest mungbean and winter wheat grain yield and gross margins compared to conventional fallow and the added stubble. Cash crop yields were related to cover crop biomass production, biomass N accumulation, residue carbon to N ratio, and legacy impacts through effects on soil water availability at cash crop sowing. Given the additional grain yield and gross margin benefits following grass-associated cover crops, they may provide a potential alternative fallow soil water and N management option that could improve crop productivity and cropping system resilience in water-limited environments.

**Keywords:** crop rotation; diversification; ecosystem services; water-limited environment

## 1. Introduction

In dryland agroecosystems, cropping system productivity and profitability are often driven by soil water and nitrogen (N) availability, and understanding these drivers is critical for sustainable and stable yields [1–5]. Conventional management in a typical dryland crop rotation involves the use of fallow periods to recharge soil water and N for the subsequent cash crop [6,7]. This reduces the risk of crop failure and thus improves the stability of crop production. However, recent literature has shown that fallow efficiency and N accumulation in conventional fallows are low, with evaporative water losses of up to 60% [8]; in addition, applied N losses of 26–37% have been reported during fallows [9,10]. This necessitates the development of more sustainable and profitable alternative practices that can enhance both fallow efficiency and N accumulation while maintaining high crop yields.

One such practice is the use of cover crops, which are non-harvested crops grown during a fallow period. Cover crops can be used to enhance fallow efficiency and N accumulation, provide a disease and pest break, and potentially improve farming system productivity [11,12]. The use of cover crops has been considered an important component of crop diversification that could improve cropping sustainability and resilience. For example, in a semi-arid cropping system, replacing fallow periods with spring oat (*Avena sativa* L.), spring triticale (×*Triticosecale* Wittmack.), spring pea (*Pisum sativum* L.), buckwheat (*Fagopyrum esculentum* L.), turnip (*Brassica campestris* L.), forage radish (*Raphanus sativus* L.), or a mixture of these species resulted in an increased precipitation storage efficiency of 38–146% compared with fallow [5]. In a Mediterranean climate, Li et al. [13] reported higher maize and tomato yield stability following cover crops than under conventional systems. And in a recent meta-analysis, cover crops were found to be able to increase cash crop yields by up to 15% depending on pedo-climactic factors [14].

Despite the high number of studies demonstrating the potential of cover crops to provide a wide range of ecosystem service benefits, on-farm adoption of cover cropping in water-limited environments has been limited by perceived potential soil water and N management costs and subsequent yield penalties [15–17]. This may be because dryland cropping systems are primarily driven by soil water availability, and decisions on whether to fallow or cover crop can result in large differences in cash crop productivity and overall farm gross margin [18]. The majority of studies finding positive effects of cover crops on crop productivity and profitability are from high precipitation regions (>700 mm annual precipitation), with highly inconsistent results in drier regions [14,19]. Furthermore, there is limited knowledge on the extent to which cover crops can alter fallow soil water and mineral N without impairing crop yield or farm gross margin. Thus, there is need to better understand cover crop management implications on fallow soil water and mineral N, as well as their legacy impact on subsequent cash crop yields and profitability.

Despite increasing interest in using cover crops for fallow management, there have been limited studies investigating cover crop selection according to a target agroecosystem service/function or suite of services/functions. Many studies have posited that using cover crops 'in general', i.e., encompassing divergent functional traits, can enhance multiple ecosystem functions, including both N cycling and water conservation, and improve on-farm diversity for sustainable crop production. However, Zhang et al. [20] showed that using acquisitive cover crops (i.e., with high root branching density, root length density, and low C:N ratio) including brassicas, grasses, and legumes promoted higher crop productivity than conservative cover crops (i.e., with high root C:N ratio, leaf lignin content, and high leaf N content). These functional traits relate to how plants acquire and utilize environmental resources for growth and development [21,22]. Trade-offs between functional traits promoting rapid resource capture and efficient resource conservation have limited the potential of diverse cover crops to deliver multiple ecosystem services [20,23]. In addition, studies have shown uneven effects on soil water and N dynamics from the planting of different cover crops with contrasting functional traits. For example, in a semi-arid environment, mixtures of winter canola (*Brassica napus* L.), radish (*Raphanus sativus* L.), turnip, lentil (*Lens culinaris* Medik.), peas, and millet (*Pennisetum glaucum* (L.) R. Br), which encompass divergent functional traits, promoted greater soil N scavenging, but also increased subsequent cash crop water stress, reducing subsequent crop yields [7]. Therefore, further assessment is needed of whether manipulation of cover crop functional trait diversity in mixtures enhances or reduces subsequent crop productivity and profitability and whether this is mediated via fallow soil water and mineral N legacy impacts.

This study focused on understanding the impact of replacing part of the conventional dryland fallow period with diverse cover crops on fallow soil water and mineral N dynamics and the legacy impacts on subsequent cash crop productivity and profitability. The cover crops planted: Brassicaceae (forage rape—*Brassica napus* L.), Fabaceae (common vetch (*Vicia sativa* subsp. *sativa* L.) and fababean (*Vicia faba* L.)), and Poaceae (forage oat—*Avena sativa* L.) represent contrasting functional traits related to growth, phenology, and soil water

and N acquisition and use strategies. Grasses generally have high biomass production, moderate water use, high residue C/N ratio, and a fibrous rooting pattern that allows for greater soil water and N acquisition at different lateral directions. Legumes can fix N, and generally have lower biomass, water use, and residue C/N ratio than grasses. Brassicas have a deep tap rooting pattern that enables access to water and N from deep soil layers and have high biomass production and water use with a low residue C/N ratio [14,21,24]. The specific objectives of this study were to (i) examine the legacy impacts of cover crop type on subsequent cash crop productivity and profitability and (ii) determine the drivers of cover crop legacy effects on crop yields as defined by fallow soil water and mineral N dynamics. We hypothesized that cover crop mixtures incorporating contrasting functional traits can augment complementary functions that improve soil water storage and mineral N accumulation during fallow with significant positive effects on subsequent cash crop productivity and profitability. To test these hypotheses, we first highlight the roles and limitations of cover crops for fallow replacement in sustainable crop–fallow rotation towards addressing some of the barriers to cover crop adoption in a water-limited environment. Secondly, we utilized 3-site-year field experiments to quantify the legacy of cover crops with contrasting functional traits (Poaceae vs. Fabaceae vs. Brassicaceae) on subsequent cash crop productivity and profitability. Finally, we used structural equation modelling (SEM) to understand the drivers of cover crop legacy effects. This study supports effort towards addressing some of the barriers to cover crop adoption in water-limited environments by providing additional evidence required to make better cropping decisions that could ultimately result in improvement in cropping system productivity, profitability, and sustainability.

## 2. Materials and Methods

### 2.1. Experimental Sites and Field Description

A field experiment was conducted over three consecutive years in a semi-arid subtropical climatic region of the northern grain region of Australia. The site was located at The University of Queensland, Gatton Crop Research Station (27.5364° S, 152.341° E) on a black deep-cracking, self-mulching clay-rich vertosol predominantly used for cropping. Initial soil samples (0–10, 10–30, and 30–60, 60–90, 90–120 cm) were collected by compositing 11 cores (44 mm diameter) prior to cover crop planting in 2020, and the initial soil characteristics are reported in Garba et al. [24]. Briefly, the site was a black vertosol soil (Isbell 1996) with a sandy clay loam surface graduating to clay at depth and plant available water capacity (PAWC) of 215 mm for 0–120 cm soil profile depth. The top 10 cm soil layer has a neutral pH (in water), becoming slightly alkaline with depth, and a Walkley–Black organic carbon content of >1.5%.

Figure 1 shows a schematic representation of the three main categories of the fallow management system tested across the 3-site years, showing the different phases of the crop-fallow rotation, in addition to cumulative precipitation during the cover crop phase, fallow period, and cash crop phases. In Year 1, cumulative precipitation before cover crop sowing averaged 221 mm with only 41 mm during the cover crop phase. Therefore, 44 mm of supplemental irrigation was applied (quantity within 80% of the long-term precipitation average of the experimental sites) to ensure cover crop growth. In Years 2 and 3, adequate precipitation was received and therefore no supplementary irrigation was provided.

### 2.2. Treatments and Experimental Design

The winter trials comprised 12 experimental treatments including a conventional fallow (control) and a stubble application treatment (local check) (Figure 1). The stubble application treatment consisted of applying high C/N ratio (up to 64:1) barley/oat residues at a rate of 10 t ha$^{-1}$ to the soil surface. The aim of this treatment was to provide an upper extreme of groundcover and N immobilization effects on soil water and mineral N. The stubble was added at cover crop termination and maintained as a chemical fallow together with the conventional fallow until cash crop sowing each year. The cover

crop treatments included three commonly used cover crop functional types: Brassicaceae (forage rape—*Brassica napus* L.), Fabaceae (common vetch (*Vicia sativa* subsp. *Sativa* L.) in Year 1 and fababean (*Vicia faba* L.) in Year 2), and Poaceae (forage oat—*Avena sativa* L.). These were grown in monoculture (100% forage oat, 100% common vetch/fababean, 100% forage rape), in even two-species mixtures (50% forage oat: 50% common vetch/fababean, 50% forage rape: 50% common vetch/fababean, 50% forage oat: 50% forage rape), and variable three-species mixtures (an even three-way mixture, i.e., 33% forage oat: 33% common vetch/fababean: 33% forage rape, and dominant mixtures, i.e., 70% forage oat: 15% common vetch/fababean: 15% forage rape, 15% forage oat: 70% common vetch/fababean: 15% forage rape, and 15% forage oat: 15% common vetch/fababean: 70% forage rape) (Table 1). The sowing proportions were adjusted to the standard seeding rates for monoculture of forage oats (40 kg ha$^{-1}$), forage rape (4 kg ha$^{-1}$), common vetch (40 kg ha$^{-1}$), and fababean (200 kg ha$^{-1}$). Each of the 12 treatments was replicated 4 times in a randomized complete block design on an 8 m × 5 m plot, for a total of 48 plots each year.

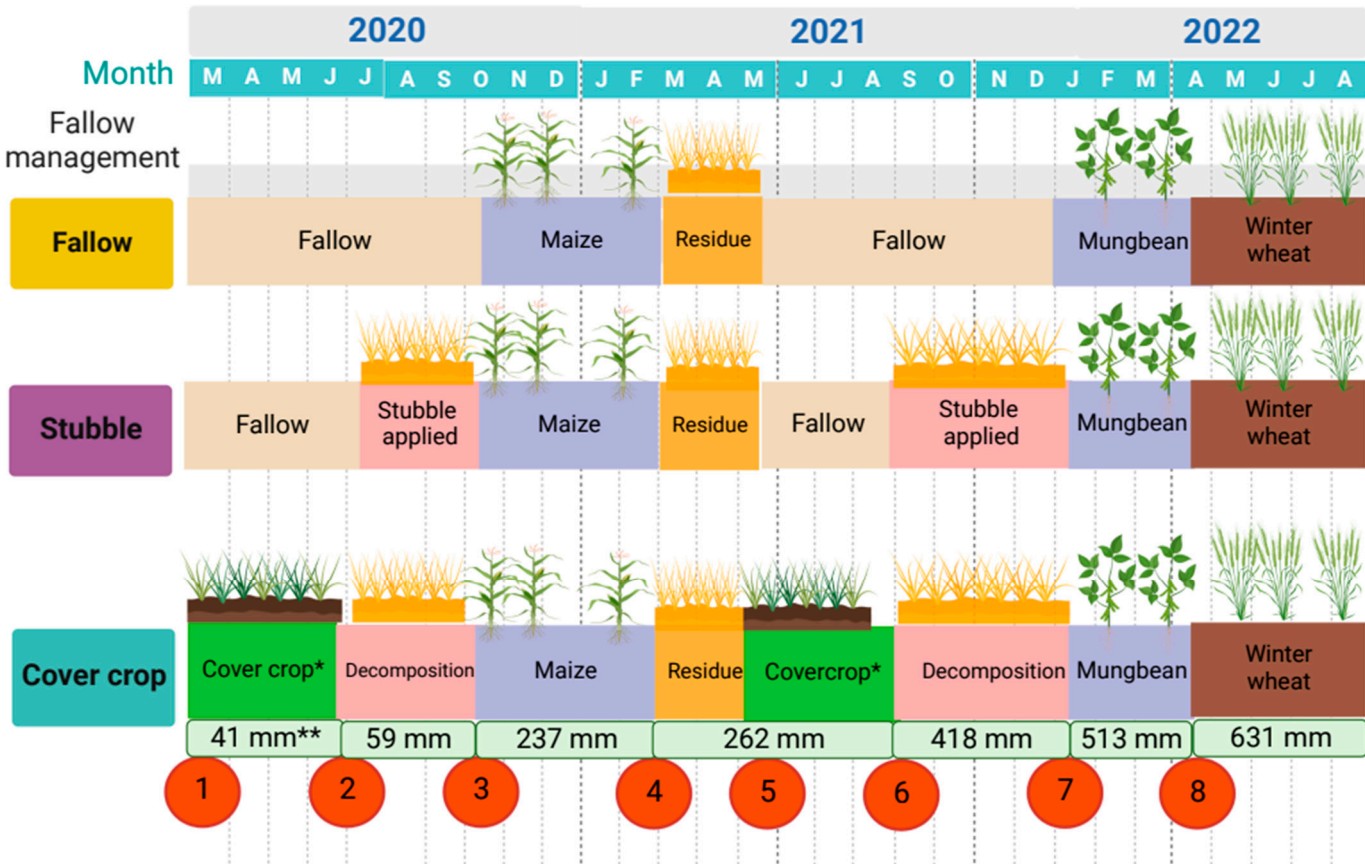

**Figure 1.** Schematic representation of the three main categories of the fallow management system tested across the 3-site years showing the different phases of the crop–fallow rotation. The red circles show the timing of soil water and mineral N samplings at cover crop sowing (1, 5), cover crop termination (2, 6), cash crop sowing (3, 8), and cash crop harvesting (4, 7). Cumulative precipitation (mm) during the different phases is provided in the bottom bar. * Cover crop treatments varied with the choice of functional types and mixture composition (Poaceae vs. Fabaceae vs. Brassicaceae). ** Only 41 mm of precipitation was received during the cover crop phase in Year 1 and was supplemented with another 44 mm to ensure crop growth, i.e., this specific cover crop phase received 85 mm of total water input, comprising rainfall and irrigation.

**Table 1.** Cover crop seeding rates (kg seeds ha$^{-1}$), under various sowing proportions. The—show that the specific cover crop functional type was not included in the given treatment.

| Treatment (GR:LG:BR) | Grass (GR) | † Legume (LG) | | Brassica (BR) |
| --- | --- | --- | --- | --- |
| | Forage Oat | Common Vetch | Fababean | Forage Rape |
| Conventional fallow | - | - | - | - |
| Stubble | - | - | - | - |
| Fallow | 40 | - | - | - |
| Stubble | - | 40 | 200 | - |
| 100:0:0 | - | - | - | 4 |
| 0:100:0 | 20 | 20 | 100 | - |
| 0:0:100 | 20 | - | - | 2 |
| 50:50:0 | - | 20 | 100 | 2 |
| 50:0:50 | 13.2 | 13.2 | 66 | 1.32 |
| 0:50:50 | 28 | 6 | 30 | 0.6 |
| 33:33:33 | 6 | 28 | 140 | 0.6 |
| 70:15:15 | 6 | 6 | 30 | 2.8 |

† Common vetch was planted in Year 1; Fababean was planted in Year 2.

*2.3. Crop Management*

Cover crops were sown using a drill planter at 35 cm row spacing after incorporation of 30 kg ha$^{-1}$ of starter GranulockZ® fertiliser containing 11% N, 21.8% P, 0% K, 4% S, 1% B (Incitec Pivot Fertilisers, Melbourne, Australia). The legume cover crops were inoculated with NoduleN™ group F inoculant (strain #WSM1455) to ensure adequate nodulation for biological N fixation. Cover crops were terminated at late stem elongation of the forage oats (90 and 80 days after sowing in Year 1 and Year 2, respectively). Cover crops were terminated with glyphosate (450 g L$^{-1}$ isopropylamine at 2.4 L ha$^{-1}$ + sharpen (700 g kg$^{-1}$ saflufenacil) + 1% *v/v* hasten adjuvant with residue left standing before cash crop sowing in the summer. In Year 1, maize (*Zea mays* L.) was seeded approximately 110 days after cover crop termination (DAT) on 81 cm row spacing with a target density of 6 plants m$^{-2}$. A basal application of urea fertiliser was applied at the rate of 120 kg N ha$^{-1}$ followed by a side dress of 60 kg N ha$^{-1}$ at the V6 growth stage. The N fertiliser rate applied was below the 90th percentile of the recommended fertiliser rate for the region to allow for an understanding of the impact of the cover crops on cash crop N dynamics. In Year 2, mungbean (*Vigna radiata* L.) was sown two months later. Mungbean was inoculated with group I (strain #CB1015) inoculant before planting and sown on 50 cm row spacing at a target density of 30 plants m$^{-2}$. After the mungbean harvest (Year 3), we opted to double crop with winter wheat (*Triticum aestivum* L.) instead of cover crop. This decision was taken to match grower practice in the region in response to ample rainfall. In all years, cash crops were harvested at physiological maturity. The timing of agronomic operations is presented in Table 2.

*2.4. Soil Water and Nitrogen Measurements*

In addition to the baseline soil samples taken at the beginning of the trial, soil samples were taken at cover crop sowing, cover crop termination, cash crop sowing, and cash crop harvest from each plot for the determination of plant available water content (PAW) and soil mineral N (SMN), as shown in Figure 1. Two soil cores were collected per plot to 120 cm depth using a hydraulic corer (44 mm diameter). Cores were sectioned into 0–10, 10–30, 30–60, 60–90, and 90–120 cm depth soil layers and combined based on the soil depth layer to give a composite set of samples per plot. A sub-sample was taken and oven-dried in a forced-air oven at 105 °C until constant weight to determine gravimetric soil water content (GWC). The GWC was adjusted to volumetric water content (VWC) by multiplying GWC with soil layer bulk density. In addition, the permanent wilting point was determined as the lower limit soil water at 15 bar pressure (LL15) using a Ceramic Pressure Plate Extractor (Agro-Ecosystems Soil Management Solutions). For each plot, PAW at each sampling event was determined as the difference between VWC and LL15. Net soil water accumulation was

determined as the proportion of precipitation that fell between cover crop termination and cash crop sowing and that was stored in the soil. Fallow efficiency (%) was calculated as the proportion of fallow precipitation that was converted to PAW at cash crop sowing (SWP). For the determination of plant available N (mineral N), sub-samples of the composite samples of each plot were dried at 40 °C until constant weight and ground through a 2-mm sieve. Ground samples were analysed for $NO_3$-N and $NH_4$–N concentrations at a commercial laboratory by colourimetric assay after extraction with 30 mL 2 M KCl [25]. Mineral N was determined as the sum of $NO_3$-N and $NH_4$–N concentrations and converted to per ha quantities by adjusting for soil layer thickness and bulk density. Net soil mineral N accumulation was determined as the apparent N mineralised from cover crop residue between cover crop termination to cash crop sowing. Summary values of soil water and mineral N across the two-year cover crops are presented in Table 3. Full methodological details and complete soil data are presented in Garba et al. [26,27].

**Table 2.** Agronomic management for crop sowing, cover crop termination, fertilization, and harvesting.

| | Management | Rate (kg ha$^{-1}$) | Date | Method |
|---|---|---|---|---|
| | | | Year 1 | |
| 1 | GranulockZ fertiliser | 30 | 20 March 2020 | Banded |
| 2 | Cover crop sowing | | 23 March 2020 | Drill |
| 3 | Cover crop termination | | 23 June 2020 | Chemical: Glyphosate at 2.4 L ha$^{-1}$ + sharpen (700 g kg$^{-1}$ saflufenacil) plus 1% *v/v* hasten adjuvant |
| 3 | Basal urea application | 120 | 18 October 2020 | Banded |
| 5 | Cash crop sowing | | 20 October 2020 | Mechanical |
| 6 | Top dress urea | 60 | 14 November 2020 | Banded |
| 7 | Cash crop harvest | | 20 February 2021 | Drill |
| | | | Year 2 | |
| 8 | Cover crop sowing | | 25 May 2021 | Drill |
| 9 | Cover crop termination | | 16 August 2021 | Chemical: Glyphosate at 2.4 L ha$^{-1}$ and sharpen (700 g kg$^{-1}$ saflufenacil) plus 1% *v/v* hasten adjuvant |
| 10 | Basal urea application | 120 | 02 November 2021 | Banded |
| 11 | Cash crop sowing | | 13 January 2022 | Drill |
| 12 | Cash crop harvest | | 12 April 2022 | Mechanical |
| | | | Year 3 | |
| | Pre-sowing herbicide | | 27 May 2022 | Chemical: Glyphosate at 2.4 L ha$^{-1}$ and sharpen (700 g kg$^{-1}$ saflufenacil) plus 1% *v/v* hasten adjuvant |
| | Cash crop sowing | | 02 June 2022 | Drill |
| | Cash crop harvest | | 10 November 2022 | Mechanical |

### 2.5. Crop Measurements

At cover crop termination, biomass was sampled from a 1 m$^2$ quadrat prior to herbicide kill. Biomass was partitioned into cover crop species for each plot and then oven-dried at 65 °C until constant weight to determine dry matter accumulation. Dried samples were ground through a 2 mm sieve and analysed for total C and N concentration by Dumas combustion and analysis in a LECO C-N analyser (CN 928 Series, LECO Corporation, Geleen, The Netherlands). In addition, sub-samples of legume ground plant materials were further ground to 0.5 mm and analysed for biologically fixed N (BFN) at the Stable Isotope Facility, University of California, Davis, using a PDZ Europa ANCA-GSL elemental analyser interfaced to a PDZ Europa 20–20 continuous isotope mass spectrometer (IRMS) (Sercon Ltd., Cheshire, UK). The BFN was determined by the natural abundance method using forage oat as the reference plant. The $\delta^{15}N$ of the legume and the forage oats were used to estimate the total BFN using the procedure of Unkovich et al. [28]. Summary values of cover crop performance are presented in Table 4 (see Garba et al. [26] for full details).

**Table 3.** Distribution of soil water and mineral N dynamics and percentage of cover crop residue remaining at cash crop sowing as affected by cover crop functional type and mixture sowing proportions of oats, common vetch/fababean, and forage rape averaged across the two years of cover cropping.

| Treatment (GR:LG:BR) | PAW at Cash Crop Sowing (mm) | | | | Fallow Efficiency (%) | | | | SMN at the Cash Crop Sowing (kg N ha$^{-1}$) | | | | Net Soil Mineral N Mineralization between Cover Crop Termination and Cash Crop Sowing (kg N ha$^{-1}$) | | | | Residue Remaining at Cash Crop Sowing (%) | | | |
|---|---|---|---|---|---|---|---|---|---|---|---|---|---|---|---|---|---|---|---|---|
| | † Min | β Max | Mean | Φ CV | Min | Max | Mean | CV | Min | Max | Mean | CV | Min | Max | Mean | CV | Min | Max | Mean | CV |
| Fallow | 121 | 175 | 154 | 13.3 | 32 | 46 | 40 | 12.3 | 171 | 219 | 197 | 8.8 | 36 | 64 | 47 | 20.6 | | | | |
| Stubble | 129 | 198 | 159 | 13.1 | 26 | 52 | 40 | 19.2 | 152 | 299 | 202 | 23.6 | 1 | 176 | 55 | 100.7 | 18 | 64 | 40 | 50.4 |
| 100:0:0 | 73 | 174 | 130 | 29.3 | 12 | 41 | 27 | 33.8 | 121 | 222 | 161 | 20.7 | 43 | 120 | 70 | 36.1 | 10 | 50 | 24 | 57.9 |
| 0:100:0 | 87 | 152 | 120 | 22.8 | 18 | 36 | 26 | 25.6 | 148 | 233 | 178 | 18.0 | 32 | 120 | 65 | 48.0 | 1 | 25 | 16 | 50.8 |
| 0:0:100 | 60 | 174 | 111 | 44.5 | 7 | 42 | 22 | 67.3 | 117 | 211 | 153 | 22.0 | 15 | 119 | 65 | 47.6 | 4 | 33 | 13 | 77.3 |
| 50:50:0 | 88 | 174 | 126 | 25.7 | 18 | 40 | 27 | 30.3 | 110 | 218 | 162 | 22.3 | 24 | 100 | 65 | 40.5 | 10 | 29 | 21 | 38.0 |
| 50:0:50 | 50 | 172 | 122 | 36.4 | 2 | 48 | 26 | 53.6 | 135 | 215 | 183 | 13.9 | 48 | 147 | 78 | 43.9 | 4 | 26 | 13 | 56.9 |
| 0:50:50 | 73 | 150 | 114 | 29.8 | 12 | 38 | 24 | 43.5 | 94 | 276 | 167 | 38.7 | 3 | 174 | 70 | 93.8 | 9 | 36 | 16 | 58.1 |
| 33:33:33 | 68 | 162 | 120 | 31.8 | 10 | 41 | 28 | 42.7 | 117 | 184 | 157 | 16.5 | 30 | 106 | 73 | 38.2 | 5 | 40 | 21 | 71.6 |
| 70:15:15 | 64 | 178 | 123 | 31.9 | 8 | 35 | 24 | 35.5 | 125 | 236 | 185 | 20.2 | 12 | 108 | 81 | 40.8 | 7 | 27 | 17 | 39.2 |
| 15:70:15 | 89 | 168 | 127 | 21.1 | 18 | 36 | 29 | 23.7 | 119 | 187 | 148 | 13.9 | 23 | 82 | 54 | 41.6 | 9 | 26 | 18 | 33.8 |
| 15:15:70 | 61 | 173 | 114 | 38.9 | 5 | 45 | 22 | 68.1 | 83 | 270 | 165 | 33.9 | 4 | 194 | 66 | 91.0 | 7 | 36 | 17 | 57.2 |

† Min = Minimum; β Max = Maximum; Φ CV = Coefficient of variation (%). PAW = Plant available water content (mm); SMN = Soil mineral N (kg N ha$^{-1}$). The ratios show the sowing proportion (GR:LG:BR) of the different cover crop functional types GR = grass; LG = legume; BR = brassica cover crops.

**Table 4.** Cover crop properties at termination as influenced by cover crop functional type and mixture sowing proportions of oats, common vetch/fababean, and forage rape averaged across the two years of cover cropping.

| Treatment (GR:LG:BR) | Aboveground Biomass (kg DM ha$^{-1}$) | | | | Shoot N Concentration (%) | | | | Biomass C/N Ratio | | | | Biomass N Retention (kg N ha$^{-1}$) | | | |
|---|---|---|---|---|---|---|---|---|---|---|---|---|---|---|---|---|
| | [†] Min | [β] Max | Mean | [Φ] CV | Min | Max | Mean | CV | Min | Max | Mean | CV | Min | Max | Mean | CV |
| Fallow | 1456 | 1816 | 1635 | 8.5 | | | | | | | | | | | | |
| Stubble | 9241 | 9281 | 9266 | 0.2 | 0.6 | 2.4 | 1.5 | 59.1 | 17.2 | 69.1 | 41.5 | 60.5 | 57.8 | 222.7 | 137.9 | 59.1 |
| 100:0:0 | 3689 | 5279 | 4532 | 13.4 | 3 | 4.1 | 3.4 | 11.6 | 9.6 | 13 | 11.4 | 10.5 | 112 | 201.1 | 155.1 | 18.1 |
| 0:100:0 | 1884 | 3654 | 2581 | 26.0 | 2.6 | 4.1 | 3.1 | 15.3 | 9.1 | 15.3 | 12.1 | 19.4 | 53.8 | 197.1 | 125.3 | 38.5 |
| 0:0:100 | 4327 | 8014 | 5755 | 20.7 | 3.5 | 4.2 | 4.0 | 6.5 | 8.2 | 10.2 | 9.1 | 6.7 | 180.4 | 297.1 | 225.9 | 17.0 |
| 50:50:0 | 3571 | 5926 | 4704 | 15.2 | 2.7 | 3.8 | 3.3 | 11.7 | 10.5 | 14.8 | 11.9 | 11.2 | 152.6 | 216.5 | 174.8 | 13.1 |
| 50:0:50 | 4003 | 7321 | 5522 | 19.9 | 3.4 | 4.5 | 3.9 | 10.0 | 8.6 | 10.5 | 9.6 | 7.9 | 148.8 | 278.2 | 212.8 | 22.1 |
| 0:50:50 | 2531 | 7474 | 4422 | 40.8 | 2.7 | 3.9 | 3.5 | 13.4 | 9.5 | 13.9 | 10.6 | 14.2 | 94.8 | 285.6 | 176.7 | 40.8 |
| 33:33:33 | 3808 | 6913 | 5509 | 19.4 | 3.2 | 4.0 | 3.6 | 7.7 | 9.3 | 11.6 | 10.3 | 7.6 | 143.4 | 253.1 | 211.1 | 19.1 |
| 70:15:15 | 3878 | 9123 | 5460 | 33.7 | 3.4 | 4.1 | 3.6 | 6.7 | 9.6 | 11.1 | 10.7 | 5.5 | 153.5 | 312.6 | 204.3 | 28.1 |
| 15:70:15 | 3227 | 5965 | 4594 | 20.6 | 2.6 | 4.0 | 3.5 | 13.8 | 9.8 | 12.6 | 10.8 | 8.3 | 140.1 | 227.4 | 171.9 | 16.2 |
| 15:15:70 | 2454 | 6705 | 4885 | 29.3 | 3.3 | 4.2 | 3.8 | 9.0 | 8.8 | 12.5 | 10 | 11.3 | 90.4 | 249.1 | 191.9 | 25 |

[†] Min = Minimum; [β] Max = Maximum; [Φ] CV = Coefficient of variation (%); C = carbon; N nitrogen; DM = dry matter. The ratios show the sowing proportion (GR:LG:BR) of the different cover crop functional types GR = grass; LG = legume; BR = brassica cover crops.

For the maize cash crop, biomass and ear samples were harvested from a 2 m$^2$ net plot and weighed in the field. A sub-sample of six ears was taken from each plot and oven dried at 65 °C until constant weight to determine shelling percentage, grain weight, and moisture content. Final maize grain yield was adjusted to 14% grain moisture content. Mungbean and winter wheat plants were harvested from 1 m$^2$ quadrats and processed to determine biomass weight, grain yields, and grain moisture. Final grain yields of mungbean and winter wheat were expressed at 13% and 12.5% grain moisture content, respectively.

Soil water (mm) at the times of cash crop sowing (*SWS*) and harvesting (*SWH*) was expressed as PAW stored in the 0–120 cm soil profile depth. Seasonal crop water use (*SWU*) was estimated as a residual of the soil water balance as expressed in Equation (1):

$$SWU \text{ (mm)} = P + I - R - D + CR - (SWH - SWS) \tag{1}$$

where *P* is precipitation (mm), *I* is supplementary irrigation (mm), *R* is runoff (mm), *D* is drainage (mm), and *CR* is the capillary rise (mm). The *D*, *R*, and *CR* parameters were assumed to be negligible because both trial sites had a slope of <2%.

The *SWU* was used together with cash crop grain yield to estimate water use efficiency (*WUE*; kg ha$^{-1}$ mm$^{-1}$) [Equation (2)].

$$WUE \left( \text{kg ha}^{-1} \text{ mm}^{-1} \right) = \frac{\text{Grain yield } \left( \text{kg ha}^{-1} \right)}{SWU \text{ (mm)}} \tag{2}$$

To ascertain the effects of cover crop residue on crop establishment, we determined the realized crop population density at harvest. In addition, crop grain protein content was determined using a calibrated near infrared whole grain analyser (CropScan 3000B; Next Instruments Pty Ltd., New South Wales, Australia).

## 2.6. Farm Gross Margin

Farm gross margin (AUD$ ha$^{-1}$) was calculated as the difference between gross income and total variable cost [Equation (3)]. Variable costs included crop seed, herbicides to terminate cover crops and control weeds, fertilisers, machinery operations, and the cost of stubble bale. Crop prices and cost of inputs were recorded for each year. Gross income was determined as the product of yield (kg ha$^{-1}$) and grain price ($ ton$^{-1}$). Data used to make these calculations are summarized in Table S1.

$$\text{Gross margin} \left( \$ \text{ ha}^{-1} \right) = \text{Gross income} \left( \$ \text{ ha}^{-1} \right) - \text{Variable cost} \left( \$ \text{ ha}^{-1} \right) \tag{3}$$

## 2.7. Data Analysis

To determine the effects of fallow management options (cover crop vs. fallow vs. stubble) on the measured variables, their effects on fallow soil water, mineral N dynamics, and legacy impact on subsequent cash crop yields, linear mixed-effects (LME) models were fitted with cover crop type as a fixed effect and replication as a random term. Models were fit using the "lme" function from the package "*nlme*" [29] in *R v4.1.0* [30] using *RStudio v1.3.959* [31]. Means separation was conducted using the Tukey HSD test [32] based on estimated marginal means using the "*emmeans*" package [33]. The experimental data were treated as cropping sequence and each year were analysed separately to account for the difference in the cash crop grown and seasonal variations.

Pearson correlation analysis was conducted to evaluate relationships between cover crop performance indicators and fallow soil water and mineral N dynamics and legacy impacts on subsequent cash crop yields. Correlation analysis was performed using the "*correlation*" package [34] with correlation plots produced using the "*corrplot*" package [35] in *R v4.1.0* [30] using *RStudio v1.3.959* [31]. The correlations were conducted using studentised residuals extracted from the full LME models to remove cover crop treatment, year, and cash crop identity effects, as these effects were already analysed in the LME analyses.

Following correlation analysis, the variables with a significant relationship with subsequent cash crop productivity parameters were used in the SEM.

A structural equation modelling (SEM) approach was utilised to model the covariances in relationships between cover crop properties and cash crop yield, as mediated through changes in fallow soil water and mineral N (Figure S1). The SEM was conducted on the same studentised residuals extracted from the full LME models. By removing the effects of cover crop treatment, cash crop identity, and year, the SEM analysed the effects of intrinsic cover crop properties without influence of cover crop and cash crop identities and differences in seasonal conditions. We excluded the fallow and the stubble treatment from this analysis because they had no input values for cover crop biomass and quality parameters. We started with a priori global model based on the hypothesis that cash crop productivity (grain yield and protein content) is influenced by cover crop performance and fallow, as well as through direct and indirect effects on fallow soil water and mineral N dynamics. The global model was then refined into a more parsimonious model by assessing the 'goodness of fit' of the model using Chi-squared ($\chi^2$) maximum likelihood; comparative fit index (*CFI*); root mean square error of approximation (RMSEA); and standardised root mean square residual (*SRMR*) [36,37]. An SEM is considered to have a good fit when *RMSEA* < 0.05; *SRMR* < 0.08; *CFI* > 0.95 and Fisher's *p*-value is $0.05 < p \leq 1.00$, based on the $\chi^2$ statistic [38]. The SEM was conducted using the "*lavaan*" package [39] implemented in "jamovi" (https://www.jamovi.org) open-access software (accessed 19 December 2022). All figures were produced using the "*ggplot2*" package [40].

## 3. Results

### 3.1. Legacy Effects of Cover Crops on Subsequent Cash Crops

The legacy impact of cover crop type on subsequent cash crop biomass, grain yields, realised plant population, grain protein content, water use efficiency, and farm gross margin are presented in Figures 2–4. In Year 1, the different fallow management options had no significant effects ($p > 0.05$) on realised maize plant density (#plants m$^{-2}$), aboveground biomass kg ha$^{-1}$), and grain protein content (%). However, maize grain yield was significantly affected ($p = 0.022$) by fallow management (Figure 2). Maize grain yield was highest in the stubble treatment (12,896 kg ha$^{-1}$) and lowest following brassica monoculture (0:0:100), averaging 6070 kg ha$^{-1}$. A similar trend was observed with water use efficiency, where the maize crop had the highest water use efficiency following stubble (44 kg ha$^{-1}$ mm$^{-1}$) and lowest following brassica monoculture (16 kg ha$^{-1}$ mm$^{-1}$). Growing cover crops instead of the conventional fallow increased total cost at the farm level (Table S1), and consequently, a significant difference ($p = 0.032$) was observed in farm gross margin in Year 1. Consequently, the stubble had a 2-fold higher gross margin than conventional fallow and cover crop treatments (Figure 2). The highest gross margin benefit was following stubble ($2278 ha$^{-1}$), 100:0:0 (grass:legume:brassica; $1145 ha$^{-1}$), fallow ($1049 ha$^{-1}$), 33:33:33 ($990 ha$^{-1}$), 50:50:0 ($913 ha$^{-1}$), and 0:100:0 ($820 ha$^{-1}$). Gross margins were significantly lower in 0:0:100 ($187 ha$^{-1}$), 50:0:50 ($315 ha$^{-1}$), 70:15:15 ($346 ha$^{-1}$), 15:15:70 ($363 ha$^{-1}$), and 0:50:50 ($372 ha$^{-1}$). Compared to conventional fallow, the 100:0:0 resulted in a +9% change in farm gross margin ($+96 ha$^{-1}$), while the most negative change in gross margins was in the order 0:0:100 (−82%; −$862 ha$^{-1}$), 50:0:50 (−70%; −$734 ha$^{-1}$), 70:15:15 (−67%; −$703 ha$^{-1}$), 15:15:70 (−65%; −$686 ha$^{-1}$), and 0:50:50 (−65%; −$677 ha$^{-1}$).

In Year 2, fallow management significantly ($p = 0.012$) affected the realised mungbean plant density, with the conventional fallow having the highest density (43 plants m$^{-2}$), while the 15:15:70 cover crop mixture had the lowest (30 plants m$^{-2}$). The monocultures and the stubble were on par regarding the realised plant density (Figure 3). Both cash crop grain and biomass yield were significantly ($p < 0.05$) influenced by fallow management legacy. The 0:50:50 cover crop mixture produced the least cash crop biomass (4053 kg ha$^{-1}$) while the 0:50:50 had the highest biomass (6060 kg ha$^{-1}$). Mungbean grain yield in Year 2 was similar between the cover crop and stubble, with the 50:0:50 cover crop mixture having the highest grain yield (2458 kg ha$^{-1}$) and the lowest following conventional fallow

(1739 kg ha$^{-1}$). A similar trend was observed with mungbean water use efficiency between the fallow and cover crop treatments, falling in the range of 3.23–4.49 kg ha$^{-1}$ mm$^{-1}$, although it was lowest following conventional fallow and highest following 50:0:50 cover crop mixture. Mungbean grain protein (%) was affected by fallow management, with the 70:15:15 mixture having the highest. A significant difference (*p* = 0.046) in farm gross margin was observed in the case of mungbean, but the average gross margin was highest following 50:0:50 ($1019 ha$^{-1}$) and 100:0:0 ($1000 ha$^{-1}$), and lowest following 0:50:50 ($327 ha$^{-1}$) and 0:100:0 ($392 ha$^{-1}$).

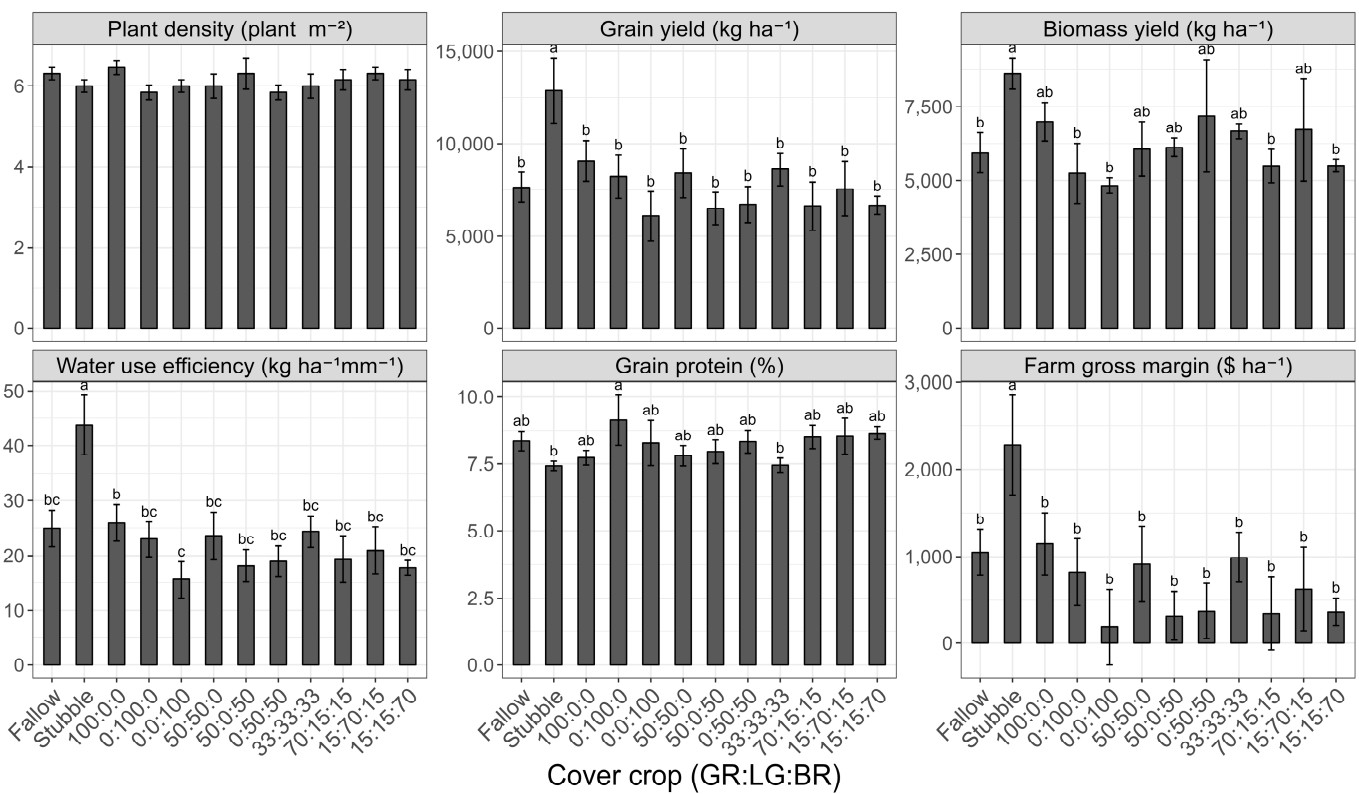

**Figure 2.** Effects of fallow management option on realised maize plant density (#plants m$^{-2}$), biomass and grain yield (kg ha$^{-1}$), grain protein (%), water use efficiency (kg mm$^{-1}$ ha$^{-1}$), and farm gross margin ($ ha$^{-1}$). Mean values followed by a different letter(s) within a column indicate a significant difference (*p* < 0.05) among treatments according to Tukey's HSD test. The ratios show the sowing proportion (GR:LG:BR) of the different cover crop functional types. GR = grass; LG = legume; BR = brassica cover crops.

　　Following two years of cover cropping, winter wheat grain yield, biomass, water use efficiency, and farm gross margin were significantly affected by fallow management (cover crops, conventional and stubble treatments) in Year 3. Average grain yield was highest following 100:0:0 (4736 kg ha$^{-1}$) and stubble (4727 kg ha$^{-1}$), while 0:100:0 had the lowest yield (3727 kg ha$^{-1}$). A similar trend was observed with aboveground biomass, where the 0:100:0 accumulated the lowest biomass (3727 kg ha$^{-1}$). Surprisingly, winter wheat water use efficiency was highest following 0:100:0 (kg ha$^{-1}$ mm$^{-1}$) while 33:33:33 was the lowest (8.0 kg ha$^{-1}$ mm$^{-1}$).

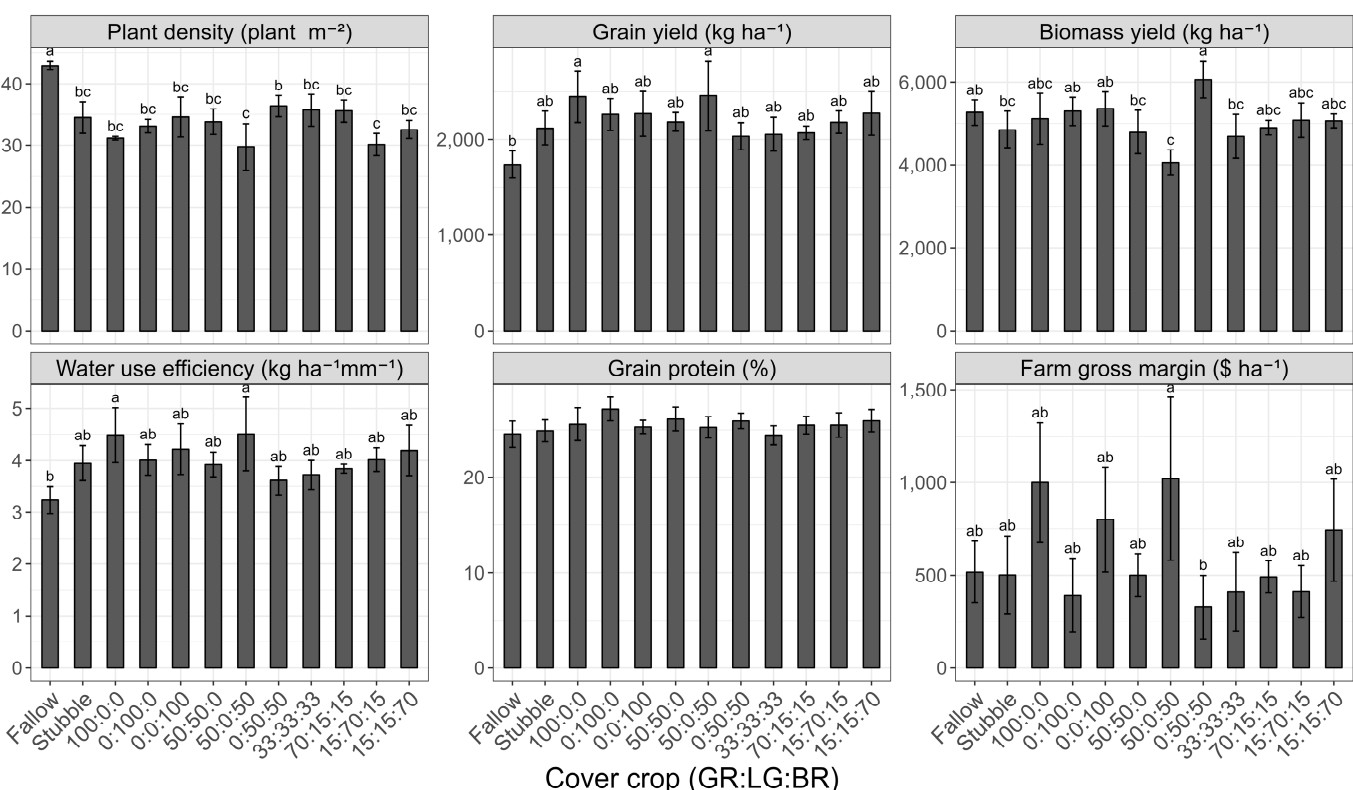

**Figure 3.** Effects of fallow management option on realised mungbean plant density (#plants m$^{-2}$), biomass and grain yield (kg ha$^{-1}$), grain protein (%),water use efficiency (kg mm$^{-1}$ ha$^{-1}$), and farm gross margin ($ ha$^{-1}$). Mean values followed by a different letter(s) within a column indicate a significant difference ($p < 0.05$) among treatments according to Tukey's HSD test. The ratios show the sowing proportion (GR:LG:BR) of the different cover crop functional types GR = grass; LG = legume; BR = brassica cover crops.

### 3.2. Relationships between Cover Crop Performance and Cash Crop Productivity

Pearson correlation coefficients (Figure 5) showed the relationships between each of the cover crop variables and the subsequent cash crop productivity parameters. Significant positive correlations were found between cash grain yield with cover crop biomass ($r = 0.37$), fallow soil water accumulation ($r = 0.32$), fallow efficiency ($r = 0.50$), and soil water content at cash crop sowing ($r = 0.49$). A negative correlation was observed between grain yield and grain protein ($r = -0.40$), while the grain yield was positively correlated with water use efficiency ($r = 0.95$) and biomass yield ($r = 0.45$). Cash crop biomass yield positively correlated with fallow efficiency ($r = 0.43$) and soil water content at cash crop sowing ($r = 0.35$). The cash crop grain protein content had a negative correlation with cover crop biomass ($r = -0.38$), N retention in biomass ($r = -0.33$), and cover crop residue remaining at cash crop sowing ($r = -0.37$). Crop water use efficiency also correlated significantly with C/N ratio ($r = 0.64$) and soil water content at cash crop sowing ($r = 0.66$). Cover crop biomass correlated significantly with C/N ratio ($r = 0.79$), N retention in biomass ($r = 0.55$), and cover crop residue remaining at cash crop sowing ($r = 0.76$).

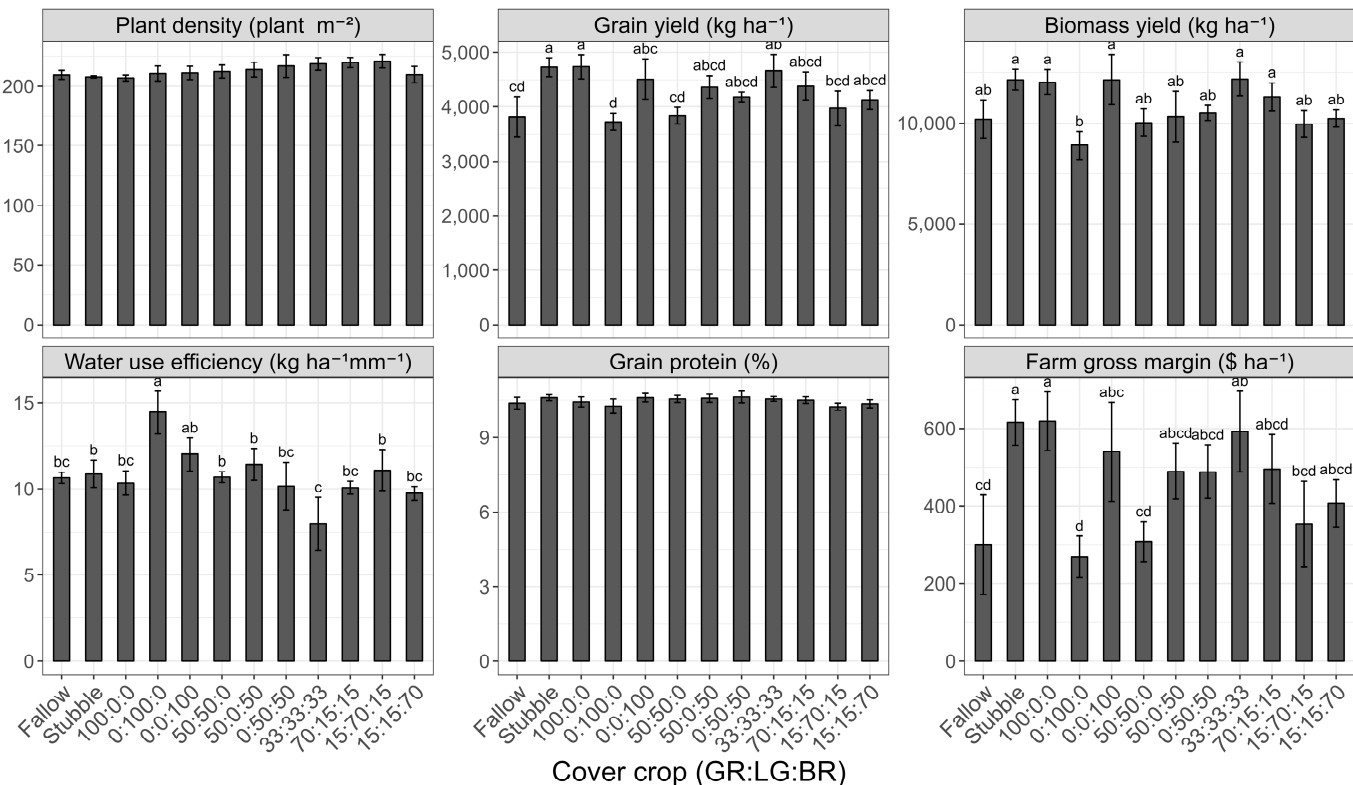

**Figure 4.** Effects of fallow management option on realized winter wheat plant density (#plants m$^{-2}$), biomass and grain yield (kg ha$^{-1}$), grain protein (%),water use efficiency (kg mm$^{-1}$ ha$^{-1}$), and farm gross margin ($ ha$^{-1}$) in Year 3. Mean values followed by a different letter(s) within a column indicate a significant difference ($p < 0.05$) among treatments according to Tukey's HSD test. The ratios show the sowing proportion (GR:LG:BR) of the different cover crop functional types GR = grass; LG = legume; BR = brassica cover crops.

The SEM analysis identified possible mechanisms underlying the observed patterns in the impact of cover crop performance on fallow soil water and N dynamics and their legacy impacts on subsequent cash crop yields (Figure 6). The SEM identified both direct and indirect effects of cover crop characteristics on fallow soil water and N at cash crop sowing and subsequent cash crop yields ($\chi^2$ = 5.31; *df* = 4; *SRMR* =0.037; *CFI* = 0.997; *RMSEA* = 0.082; *p* = 0.257). Cover crop biomass production and C/N ratio had a direct positive effect on subsequent cash crop grain yields. Cover crop biomass N accumulation (crop N uptake + biological N fixation [BNF]) had a negative effect on subsequent cash crop grain yield. In addition, cover crop biomass determined the percentage of residue remaining at cash crop sowing, which negatively influenced cash crop grain yield. Cover crop biomass C/N ratio negatively influenced soil mineral N accumulation during fallow and the mineral N stock had a marginal negative effect on total mineral N stock at cash crop sowing. Total soil water accumulation between cover crop termination and cash crop sowing directly influenced maize grain yield and indirectly via influence on total soil water content at cash crop sowing. Surprisingly, neither net soil mineral N accumulation between cover crop termination and cash crop sowing nor the total soil mineral N stock at cash crop significantly affected subsequent cash crop yields (Figure 6).

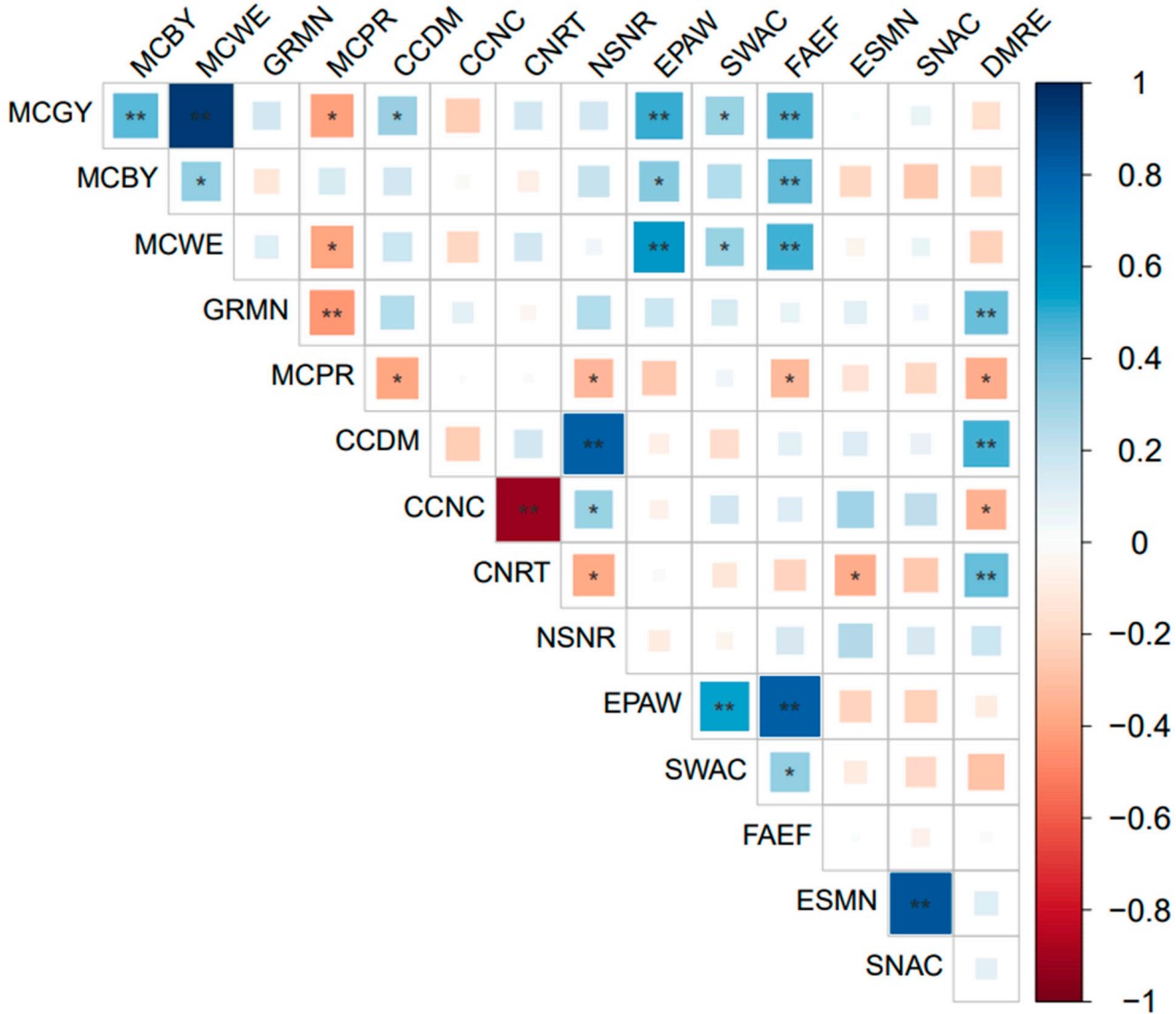

**Figure 5.** Pearson correlation coefficient (*r*) showing the relationships between cover crop performance indicators and fallow soil water and mineral N dynamics and legacy impacts on subsequent cash crop productivity and profitability. MCGY = cash crop grain yield (kg ha$^{-1}$); MCBY = cash crop biomass yield (kg ha$^{-1}$); MCWE = cash crop water use efficiency (kg ha$^{-1}$ m$^{-1}$); GRMN = farm gross margin ($ ha$^{-1}$); MCPR = cash crop grain protein (%); CCDM = cover crop biomass (kg DM ha$^{-1}$); CNCC = biomass N concentration (%); CNRT = biomass C/N ratio; NSNR = Biomass retention and supply (kg N ha$^{-1}$); EPAW = soil water at end of fallow (mm); SWAC = soil water accumulation between cover crop termination and cash crop sowing (mm); FAEF = fallow efficiency (%); ESMN = soil mineral N content at cash crop sowing (kg N ha$^{-1}$); SNAC = soil mineral N between cover crop termination and cash crop sowing (kg N ha$^{-1}$); and DMRE = cover crop residue remaining at cash crop sowing (%). * Significant at *p* < 0.05; ** Significant at *p* < 0.01.

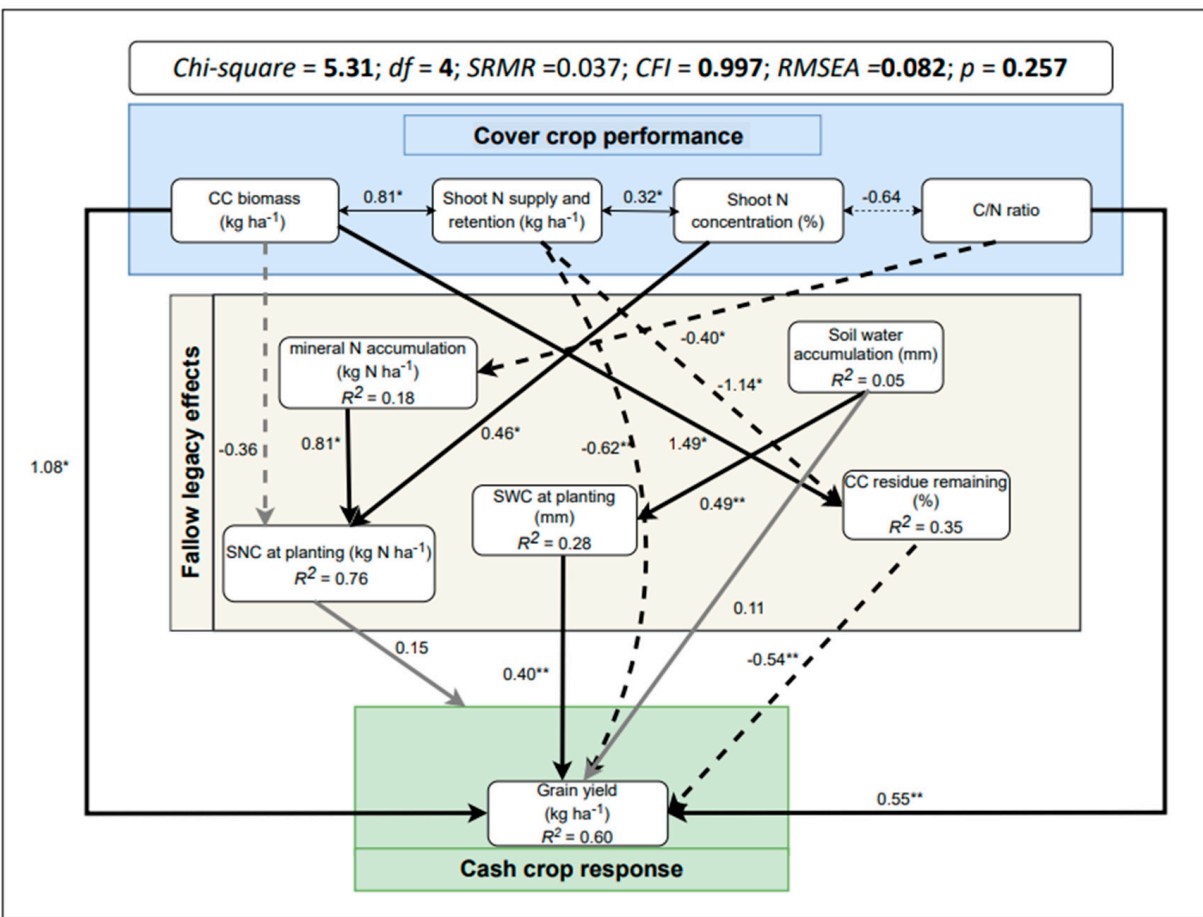

**Figure 6.** Path diagram from the structural equation model (SEM) showing relationships between cover crop variables, soil water content (SWC), soil mineral N content (SNC), and subsequent cash crop grain yields. The arrows represent significant ($p < 0.05$) unidirectional relationships between variables where solid lines show positive relationships and dashed lines show negative relationships. Standardised path coefficients are given next to each arrow. Grey lines show marginally significant ($p = 0.050$ to $0.09$) coefficients. The proportion of explained variation ($R^2$) is given for each endogenous variable. The double-ended arrows showed the correlated errors between the variables. The SEM was conducted on the residual variance from the full mixed model to remove cover crop treatment and year effects. * $p < 0.05$, ** $p < 0.01$. The full summary of the model path coefficients is provided in Table S2.

## 4. Discussion

Given the water limitations inherent in dryland agroecosystems, appropriate alternative fallow management options such as cover cropping must generate ground cover without excessive depletion of soil moisture and maintain higher soil mineral N stock at subsequent cash crop sowing. Consequently, the overall effect of the cover crops on subsequent cash crops will depend on the net effects of soil water and mineral N during the fallow. Here, we examined the legacy effects of winter cover crops on subsequent cash crop productivity and profitability in a dryland agroecosystem. We found that replacing part of the fallow period with cover crops or stubble application had differential impacts on fallow soil water and mineral N dynamics with effects on subsequent cash crop yields and gross margins, dependent on cover crop type and seasonal climatic conditions. These results are consistent with recent findings in which crop responses to cover cropping were reported to vary with genotype × environment × management interactions that modulate cover crop performance [39]. Our results show that stubble addition had the greatest impact on soil water accumulation, fallow efficiency, and subsequent cash crop yields. Brassica-associated

cover crops had the greatest biomass production and high levels of soil N retention, but consistently had low fallow soil water accumulation and overall low fallow efficiency; they were therefore associated with lower subsequent cash crop yields and gross margins. Grass- and legume-associated cover crops showed variable impacts on fallow soil water and mineral N dynamics and variable legacy effects on subsequent crops. This was mainly due to differences in their resource acquisition (soil water and N access) and use strategies (biomass accumulation, N fixation).

*4.1. Cash Crop Response to Cover Cropping*

In dryland cropping systems, available water and mineral N stored in soil during fallows are one of the key drivers of the cropping system productivity and profitability. Our results showed that the legacy impact of cover crops on subsequent cash crop yields, water use efficiency, and grain quality (protein content) varied with cover crop type and the cover crop performance. The overall effects on subsequent crop yields were highly variable and driven by complex interactions between the growing environment and management [41–44].

The high maize grain yield following stubble treatment could be due to greater persisting groundcover in the stubble treatments that maintained significantly higher soil water than all the other treatments at the end of the fallow period and had the highest fallow efficiency. Previous studies have reported higher crop yields following retained stubble [41,43,45]. Relative to conventional fallow, we found that forage oat (grass) and common vetch (legume) monocultures and the even 3-species (33:33:33; grass:legume:brassica) mixture improved maize grain yields by +19%, +8%, +10%, and +13%, while forage rape (brassica) monocultures reduced maize grain yield by −21% after one year of winter cover cropping. The high maize grain yield following legume and grass cover crops may be due to higher groundcover persistency of the grass residue and low soil water use by the legume cover crops, thus sustaining greater soil moisture retention, nutrient cycling, and improving maize grain yield. Thapa et al. [46] reported that sorghum yields were higher following oat rather than brassica cover crops due to the high C/N ratios of oat residues resulting in slower decomposition and longer groundcover residency that preserved soil water and nutrients and consequently increased yield. In the case of 3-species cover crop mixtures, the higher grain yield relative to conventional fallow may be due to increased niche complementarity in mixtures. Such complementarity allows for greater total resource capture and utilisation by the different component cover crops due to variation in their functional traits associated with soil water and N acquisition and use strategies, and plant architecture. In the variable sowing proportion mixtures, the brassica was the dominant species and consequently produced a similar yield response to the forage rape monoculture. In Minnesota, USA, Gieske et al. [47] reported a maize yield reduction with a brassica cover crop of up to 500 kg ha$^{-1}$, but the magnitude of the reduction was brassica species-specific and dependent on the impact on soil mineral N availability. Similarly, forage radish (*Raphanus sativus* L.) and 3-species mixtures of red clover (*Trifolium pratense* L.), cereal rye (*Secale cereale* L.), and forage radish reduced maize grain yield by up to 20% depending on soil water and mineral N stock at maize sowing [48].

Mungbean yields were similar across stubble and cover crop treatments, albeit slightly different from conventional fallow. This was likely due to high precipitation during both cover crop and mungbean phases that diminished potential benefits of groundcover or N mineralisation from the cover crops. This resulted in similar soil water use by the mungbean crop and therefore similar water use efficiency and grain protein content across treatments. This indicates that mungbean yield relied on soil water captured during the growing season rather than the legacy effect of the cover crop. Similar findings were reported in other semi-arid environments where the insertion of cover crops to replace a portion of the fallow periods did not change wheat yield or grain quality [5,49,50]. However, it is worth noting that these results are likely influenced by the short-term effects of cover cropping in these experiments. Thus, the differences observed in yield responses could be due to

complex interactions between cover crop characteristics and soil microbial processes that could have caused greater or lesser cycling of soil water and mineral N. This is particularly important in soils, such as vertosols found in the northern grain region of Australia, where the cropping system relies significantly on stored soil water in the profile over fallow periods to overcome large in-season rainfall variability [51].

Following two years of cover cropping, winter wheat responses to cover cropping were more dramatic, and distinct differences between different cover crop functional types were observed. Wheat grain yield was highest following grass (100:0:0) and lowest following legume (0:100:0) cover crops, suggesting that grass cover crops are consistently superior to brassica and legume cover crops in improving fallow soil water and mineral N management and consequently better cash crop outcomes. Previous studies have demonstrated that less effective groundcover from legume cover crops could exacerbate soil water loss and reduce subsequent wheat yields [52]. On the other hand, high biomass and water use from brassica cover crops increases risks of depleting soil moisture during the wheat growing season. Further long-term work is needed to unravel these mechanisms under increasing climatic variability in dryland agroecosystems.

*4.2. Drivers of Cover Crop Legacy Impacts on Subsequent Cash Crop Productivity*

The impacts of cover crops on subsequent cash crop yields represented both direct and indirect feedback effects through impacts on fallow soil water and mineral N. Consistent with previous studies (e.g., [53,54]), the SEM showed that cover crop biomass production had a direct positive effect on cash crop yields. One possible mechanism that could explain this is that cover crop biomass promoted high groundcover, thus reducing soil water evaporation and increasing water infiltration. Improved soil water infiltration and reduced evaporative loss following cover crops have been documented as the most important mechanism that drives cover crop adoption in water-limited environments [5,7,19,55]. The growing of cover crops created varying conditions for the subsequent cash crops and this drove changes in soil water and mineral N stocks and potential N release from the cover crop residue after termination. Thus, cover crop legacy effects on fallow soil water and mineral N are key determinants of subsequent cash crop productivity.

Another potential benefit of greater in-crop residue from cover crops is that the topsoil is likely to stay moist for longer and also respond in this way during in-crop rainfall. Topsoil nutrient stratification is an increasingly important issue in dryland production regions, leaving nutrients stranded when topsoil dry out [56,57]. Having greater in-crop residue cover that helps maintain longer periods of topsoil moisture may help crops access stratified nutrients for longer periods [58]. In addition, greater in-crop residue cover following cover crops may have promoted more sustained levels of N mineralisation for uptake and assimilation by the cash crop. This is particularly apparent when compared with the stubble treatment in which greater groundcover persistency led to significantly higher maize grain yield. This is consistent with the findings of Finney et al. [59] in temperate Pennsylvania, USA, where a linear relationship between maize yield and cover crop C/N ratio was reported. Surprisingly, neither the relative soil mineral N accumulation between cover crop termination and cash crop sowing nor the total soil mineral N stock at cash crop significantly influenced subsequent cash crop yields. This could be because soil mineral N stock was generally high at the sites in both years and the additional N fertiliser applied to the cash crop diminished the cash crop yield response to N. Therefore, future research is needed to understand the mechanisms for cover crop legacy impacts on subsequent cash crops and the processes (both below and aboveground) that mediate it, particularly in relation to the potential change in soil biology associated with cover crop use.

*4.3. Management and Economic Implication of Cover Crop Selection*

The adoption of cover cropping in most water-limited environments has been limited by potential soil water and N costs as well as potential yield penalties, despite strong evidence of a wide range of ecosystem services and benefits provided by cover crops. This

may be due to the fact that dryland cropping systems are driven mainly by soil water availability and decisions on whether to fallow or grow cover crops can have significant consequences on farm gross margin and return on input costs [18]. Establishing a cover crop increases total farm cost due to seed, crop establishment, and termination costs. The stubble treatment generally had a higher gross margin than the cover crop treatments in Year 1; however, applying high residue loads is not practical on a large scale in many semi-arid environments where there is generally a lack of stubble from preceding cash crops (e.g., a legume) or due to skip row configurations with uneven residue cover and exposed soils, and consequently substantial residue is not retained during the cash crop phase due to management constraints [60,61]. In contrast, the brassica cover crop consistently had a low gross margin over the three years, mainly due to its legacy effect of low cash crop yields associated with high soil water use, low C/N ratio, and poor groundcover persistency. Replacing a portion of the fallow with forage oats (100:0:0) led to significantly higher yields compared with fallow. This result indicates that grass cover crops, with their low seed cost and ease of management [62,63], could improve farm gross margins by being able to improve cash crop yields.

In cropping systems that integrate stubble retention, such as those found in semi-arid subtropical Australia, the presence of stubble is an important feature of the cropping systems due to reliance on stored moisture for crop productivity. The retained stubble enhances soil water infiltration, thus decreasing soil moisture loss via evaporation and runoff, and enhancing overall soil water storage for the subsequent cash crop [6,64]. Including cover crops during fallow periods can increase surface residue loads to further enhance the positive effects of stubble retention [8,42,65]. However, one of the challenges that high residue loads can bring is the potential impediment to farm operations and reduction in uniform seed emergence and plant density. We found that the realised mungbean plant density was higher following conventional fallow than following cover crops or stubble addition. Additional residues from either cover crops or retained stubble could have negative impacts on plant populations by delaying seed emergence and releasing phytotoxins from decomposing litter that depress crop growth, damage seedling roots, and increase susceptibility to seed-borne diseases [66–68]. Therefore, careful residue management must be considered in designing crop–fallow rotations that integrate cover crops.

Legume seeds, particularly fababean, are relatively expensive, and if BNF and groundcover benefits are to be derived from them, they need to be sown at high density. This may limit their use as fallow replacement. Nevertheless, Guldan et al. [69] found that hairy vetch and alfalfa cover crops resulted in N fertiliser replacement benefits of up to 140 kg N ha$^{-1}$, which could provide an incentive for their adoption, as reductions in N fertiliser application rates could offset their higher seed cost. In a wheat–sorghum–soybean rotation, growing late-maturing soybean cover crops has been reported to increase N fertiliser replacement value in sorghum by 51 kg N ha$^{-1}$ [70]. We did not find a significant N fertiliser replacement benefit from adding common vetch or fababean cover crops. This may partly be due to the mineralisation of their residues, which would have resulted in the rapid release of N into the mineral N pool before cash crop uptake and assimilation; this N may thus have been potentially lost by denitrification and/or leaching. However, this result could also be due to a higher soil mineral N stock (~200 kg N ha$^{-1}$) at cover crop sowing observed in the first year. This highlights the need for optimising cover crop management to maximise synchrony between cover crop residue N release and cash crop N uptake [12].

Another potential driver of the variation in the farm gross margins between years could be the differences in annual precipitation that caused variation in crop yields and thus in gross revenue. Hence, understanding factors that influence cover crop performance and their legacy impact on subsequent cash crops is critical to designing cover crop management practices that could result in long-term improvement in cropping system productivity and profitability.

## 5. Conclusions

Cover crops are one of the major components of cropping system diversification. However, the potential costs and management implications associated with soil water and N on subsequent crop yields have limited their adoption in water-limited environments. We found that replacing a portion of the conventional fallow with cover crops reduced soil water availability and mineral N at cash crop sowing; however, the legacy impact on subsequent cash crop yield and gross margin was cover crop type specific. Forage oat-associated cover crops generally had higher grain yield and profit than brassica- or legume-associated cover crops. High cash crop yields were related to cover crop biomass production, N accumulation in biomass, cover crop residue C/N ratio, and legacy impacts via effects on soil water availability at cash crop sowing. Surprisingly, neither net soil mineral N accumulation between cover crop termination and cash crop sowing nor the total soil mineral N stock at cash crop significantly affected subsequent cash crop yields. Given the additional grain yield and gross margin benefits following grass-associated cover crops, they may provide a potential alternative fallow soil water and N management option that could improve crop productivity and cropping system resilience in water-limited environments. Further research is needed to understand the mechanisms for cover crop legacy impacts on subsequent cash crops and the processes (both below and aboveground) that mediate it, particularly in relation to potential changes in soil biology resulting from the long-term use of cover crops.

**Supplementary Materials:** The following supporting information can be downloaded at: https://www.mdpi.com/article/10.3390/agronomy13010271/s1, Figure S1: The a priori conceptual framework articulating the proposed causal influences of cover crops on subsequent cash crop grain and protein yield based on the hypothesis that cash crop productivity (grain yield and protein content) is influenced by cover crop performance and fallow, as well as through direct and indirect effects on fallow soil water (SWC) and mineral N dynamics (SNC); Table S1: Price and costs of commodities, stubble, seeds, operations, and chemicals used for the calculation of farm gross margin; Table S2: Parameter estimates from the final SEM model.

**Author Contributions:** Conceptualization, I.I.G. and A.W.; methodology, I.I.G. and A.W.; formal analysis, I.I.G. and A.W.; investigation, I.I.G. and A.W.; resources, A.W.; data curation, I.I.G.; writing—original draft preparation, I.I.G.; writing—review and editing, I.I.G. and A.W.; visualization, I.I.G. and A.W.; supervision, A.W.; project administration, A.W.; funding acquisition, A.W. All authors have read and agreed to the published version of the manuscript.

**Funding:** This research received no external funding.

**Data Availability Statement:** The data presented in this study are available on request from the corresponding author.

**Acknowledgments:** The authors wish to thank Reni Apriani, Danqing Chu, Yusrina Yusof, Yifei Wang; Daniel Fay, and Hom BK Bahadur for help in the field, sampling, and sample processing.

**Conflicts of Interest:** The authors declare no conflict of interest.

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
