# Peer review of "Integrating Diverse Cover Crops for Fallow Replacement in a Subtropical Dryland: Implications on Subsequent Cash Crop Yield, Grain Quality, and Gross Margins"

_agronomy, doi:10.3390/agronomy13010271_

Round 1

Reviewer 1 Report (Previous Reviewer 1)

1. The Author(s) did address some of my comments. However, my first comment requesting the Author(s) to indicate at the end of Introduction of how the paper is structured was not addressed. To address this, Author(s) need to indicate the structure of the paper in terms of Sections e.g. Section 1 which is Introduction, might highlight the role or significant of the study in relation to what is being investigated e.g. in this study it might be the role of Cover crops in cropping systems. In this Introduction Section the study can draw from literature the general effect of cover crops on agroecosystems and further highlight the importance of the study using recent literature to fill the gap in scientific literature. Section 2 will be on Methodology and you must indicate what does it present. You continue with other Sections e.g Section 3, etc. and Section 3 is normally about findings of the study. To understand what the structure of the paper all about, the Author(s) need to refer to other Agronomy published research papers about how to present the structure of the paper.

2. The Methodology and the Results now include Year 3 in the Revised Manuscript. However, it is difficult to see in the Results Section how the results from Year 3 are presented in the Tables. In previous version of the Manuscript the Author(s) will indicate accross 2 years but in the revised Manuscript the Author(s) are no longer specific but they just stated "across years" instead of across three years. The results from the Year 3 must be clearly stated and indicated in the Tables for the benefit of the Readers and to avoid confusion to the Readers. Author(s) need to address this.

Author Response

Response to Reviewer 1

1. The Author(s) did address some of my comments. However, my first comment requesting the Author(s) to indicate at the end of Introduction of how the paper is structured was not addressed. To address this, Author(s) need to indicate the structure of the paper in terms of Sections e.g. Section 1 which is Introduction, might highlight the role or significant of the study in relation to what is being investigated e.g. in this study it might be the role of Cover crops in cropping systems. In this Introduction Section the study can draw from literature the general effect of cover crops on agroecosystems and further highlight the importance of the study using recent literature to fill the gap in scientific literature. Section 2 will be on Methodology and you must indicate what does it present. You continue with other Sections e.g Section 3, etc. and Section 3 is normally about findings of the study. To understand what the structure of the paper all about, the Author(s) need to refer to other Agronomy published research papers about how to present the structure of the paper.

Thank you for reviewing our manuscript again.

We have now provided the structure of the manuscript in lines 112 – 122 by adding the following text:

“To test these hypotheses, we first highlight the roles and limitations of cover crops for fallow replacement in sustainable crop-fallow rotation towards addressing some of the barriers to cover crop adoption in a water-limited environment. Secondly, we utilized 3-site-year field experiments to quantify the legacy of cover crops with contrasting functional traits (Poaceae vs. Fabaceae vs. Brassicaceae) on subsequent cash crop productivity and profitability. Finally, we used structural equation modelling (SEM) to understand the drivers of cover crop legacy effects. This study support effort towards addressing some of the barriers to cover crop adoption in water-limited environments by providing additional evidence required to make better cropping decisions that could ultimately result in improvement in cropping system productivity, profitability, and sustainability”

2. The Methodology and the Results now include Year 3 in the Revised Manuscript. However, it is difficult to see in the Results Section how the results from Year 3 are presented in the Tables. In previous version of the Manuscript the Author(s) will indicate accross 2 years but in the revised Manuscript the Author(s) are no longer specific but they just stated "across years" instead of across three years. The results from the Year 3 must be clearly stated and indicated in the Tables for the benefit of the Readers and to avoid confusion to the Readers. Author(s) need to address this.

Thank you for this observation. We have now included the ‘year’ term for the winter wheat, which was added in lines 413, and 439. The results of the year-3 winter wheat were included in Figure 4. We have provided in lines 194 - 199 and Figure 1 that cover crops were grown only in the first two years of the crop sequence. In year 3, we opted to double crops (mungbean and winter wheat) instead of another winter cover crop to align with the cropping practice of the region and along with our stated hypothesis. This is because we received above long-term average precipitation in this year and growing cover crops will likely reduce farm profit since we did not harvest for fodder or grow them until maturity. We have added additional clarification where we mentioned ‘across years’ in lines 230, 266-268, and 321-322.  This form Tables 3 & 4.

Reviewer 2 Report (Previous Reviewer 2)

I am happy with the changes that were made with regards to my previous comments, and thank the authors for their care and attention in their edits.

Author Response

I am happy with the changes that were made with regards to my previous comments and thank the authors for their care and attention in their edits.

Thank you for reviewing our manuscript again.

Reviewer 3 Report (New Reviewer)

Dear Authors:

1. Please let me know about all of the treatments clearly.

2. I can't find any matter about soil organic. 3. Did you measure any traits related aforementioned matter?

3. What is missing data in table 1?

4. I didn't find any explanation about tables 3, and 4!

5. I didn't understand lines 447 and 448! 

6. figure 1 is vague! I didn't find year 3! what about numbers 1 to 8 below of this schematic diagram?

Author Response

Thank you for reviewing our manuscript again.

  1. Please let me know about all of the treatments clearly.

We tested 12 treatments:

  • conventional fallow (control)
  • Stubble addition (local check)
  • grass monoculture
  • legume monoculture
  • brassica monoculture
  • 50 % grass + 50% legume mixtures
  • 50 % grass + 50% brassica mixtures
  • 50 % legume + 50% brassica mixtures
  • 33 % grass + 33 % legume + 33 % brassica mixtures
  • 70 % grass + 15 % legume + 15 % brassica mixtures
  • 15 % grass + 70 % legume + 15 % brassica mixtures
  • 15 % grass + 15 % legume + 70 % brassica mixtures

We have described this in full in lines 165-172. The additional key description was provided in Table 1 (lines 177-178) and in all figure captions were applicable.

  1. I can't find any matter about soil organic. 3. Did you measure any traits related aforementioned matter?

We have added the baseline soil carbon content in lines 134-135. Based on our review of the literature, we found many studies showed cover cropping did not induce significant short-term change in soil carbon, we, therefore, did not measure cover cropping after the first two years with cover crops. This is ongoing research; we hope to analyze this in the subsequent years.

  1. What is missing data in table 1?

The missing value (–) showed that the specific cover crop functional type was not included in the treatment. We have provided this additional information in lines 177-178.

  1. I didn't find any explanation about tables 3, and 4!

Tables 3 and 4 showed the cover crops and soil data for the first two years of cover cropping. The results of this have been published by Garba et al. [26]. Because this data has already been published, we here only provided the descriptive statistics of the cover crop data to inform readers of the range of cover crop variables. We did not attempt to analyze the covariation of the cover crop itself to avoid plagiarising the already published results.

The [26] focussed solely on cover crop properties, whereas this manuscript focuses on cash crop yields and the legacy effects of the cover crops. We have provided the summarised soil and cover crop data in these tables to provide readers with ‘quick access’. However, we understand and agree with your comment, and consequently, on lines 229-232, and 283-286 we have added text that directs readers to the recent publication [26] that provides complete details for all of these parameters, including year-by-year breakdowns:

[26] Garba, I.I.; Fay, D.; Apriani, R.; Yusof, D.Y.P.; Chu, D.; Williams, A. Fallow replacement cover crops impact soil water and nitrogen dynamics in a semi-arid sub-tropical environment. Agric. Ecosyst. Environ. 2022, 338, 108052, doi:10.1016/j.agee.2022.108052.

  1. I didn't understand lines 447 and 448! 

Thank you for this observation. We have corrected this accordingly and show the correct interpretation of the relationship as correlation (r) between cash crop yields and cover crop characteristics. See lines 446 – 457.

  1. figure 1 is vague! I didn't find year 3! what about numbers 1 to 8 below of this schematic diagram?

We have updated the figure 1 caption and provided key interpretations of the different items. We showed different phases of the cropping sequence and the key groups of treatments: conventional fallow (1), stubble addition (1), and cover crops (10 levels) resulting in a total of 12 treatments. We added these changes in lines 150-154.

Reviewer 4 Report (Previous Reviewer 5)

Well done. Compared with the previous version, the authors make very large revisions, leaps in quality, and can be accepted and published. 

Author Response

Well done. Compared with the previous version, the authors make very large revisions, leaps in quality, and can be accepted and published. 

Thank you for reviewing our manuscript again.

This manuscript is a resubmission of an earlier submission. The following is a list of the peer review reports and author responses from that submission.

Round 1

Reviewer 1 Report

General Comment: The research study titled "Effects of diverse cover crops on cash crop yield, grain quality, and gross margins in a subtropical dryland" has potential to contribute significant information that will assist in mitigation of climate change using cropping systems as part of agronomic practices. This manuscript is well written and will be of great assistance to farmers, extension officers. academics, researchers, policy makers, students, etc. To improve further the quality of the manuscript I suggest the following:

Introduction:

1. Indicate at the end of the Introduction as the last Paragraph, how the paper is structured.

Materials and Methods:

2. Line 110, start with Latitude but not with Longitude to indicate the location of the research site.

3. Line 111 on Vertisols, just give one line of major characteristics of these soils for benefit of readers who are not in soil science field.

4. Line 171, you must change 6.6 plants to 6 plants as plant cannot be a fraction.

Results:

4. Indicate level of significance (where there was significant effects or difference) in 334, 336, etc.  as you did in situations where there was no significant difference or significant effects.

Discussions:

5. Line 492 on Further research, this sentence must be moved to Conclusions as part of future research. 

Author Response

General Comment: The research study titled "Effects of diverse cover crops on cash crop yield, grain quality, and gross margins in a subtropical dryland" has potential to contribute significant information that will assist in mitigation of climate change using cropping systems as part of agronomic practices. This manuscript is well written and will be of great assistance to farmers, extension officers. academics, researchers, policy makers, students, etc.

We appreciate the reviewers’ comments we have incorporated this recommendation into the manuscript.

To improve further the quality of the manuscript I suggest the following:

Introduction:

  1. Indicate at the end of the Introduction as the last Paragraph, how the paper is structured.

We have provided the hypotheses, and the specific objectives of the study as part of the last part of the introduction section. This now provides a clear structure for the paper.

Materials and Methods:

  1. Line 110, start with Latitude but not with Longitude to indicate the location of the research site.

We interchanged the Lat – Long as recommended. See line 122

  1. Line 111 on Vertisols, just give one line of major characteristics of these soils for benefit of readers who are not in soil science field.

We have described the vertosol as “black deep-cracking, self-mulching clay-rich vertosols” See line 123

  1. Line 171, you must change 6.6 plants to 6 plants as plant cannot be a fraction.

The plant density has been changed to 6 plants m–2 See line 184.

Results:

  1. Indicate level of significance (where there was significant effects or difference) in 334, 336, etc.  as you did in situations where there was no significant difference or significant effects.

We have provided the “p-value” accordingly. See lines 338-376

Discussions:

  1. Line 492 on Further research, this sentence must be moved to Conclusions as part of future research. 

The sentence has been moved to the conclusion section. See lines 596-599

Reviewer 2 Report

Please see attached PDF for comments.

Reviewer 3 Report

The submitted manuscript "Effects of diverse cover crops on cash crop yield, grain quality, and gross margins in a subtropical dryland" is an original research on the partial replacement of conventional fallow period by cover crops. The paper investigated the effects of three types of cover crops (associated with grasses, legumes and brassicas) with different proportions on the moisture and mineral nitrogen dynamics of fallow soils, and the legacy effects on yields, grain quality and gross margins of subsequent cash crops (maize and mungbean). This study is a valuable guide to sustainable agricultural production. Moreover, the strategy used by the authors in the statistical analysis is interesting. However, I have some doubts and suggestions about the publication of the paper. In any case, I strongly encourage the authors to review it carefully point by point to clarify some issues and eventually improve the manuscript.

Some improvements are needed in the MS
1. As the authors mention in the introduction, there are a number of different types of cover crops. However, I have not seen any mention of the reasons for choosing these three types of cover crops in the paper. Could the authors add to the section "Introduction" or "Materials and Methods" the reasons for choosing these three types of cover crops as experimental materials?

2. Please check the hyphen in lines 111-116. "" and "-" are mixed.

3. Note the spaces between the numbers and units in line 168.

4. The abbreviations in Figure 1 and table 3 should be indicated in the legend. e.g. "Before CC, PAW and SMN"

5. Could Table 2 be placed in the supplementary material?

6. In lines 175-177, the author stated that "In Year 2, maize was sown 120 DAT; however, due to heavy flooding, the crop was completely damaged and mungbean (Vigna radiata L.) was sown two months later.) was sown two months later. My understanding is that the authors intended to measure the impact on maize yield, quality and gross profit in year 2, but switched to mungbean due to the impact of the extreme event. In that case, were the three types of cover crops used as experimental material chosen primarily because of their favorable effects on maize?

7. " However" has appeared several times in the manuscript. In view of the richness of the language, I suggest replacing "A" in lines 464 and 542 with " Nevertheless".

8. The conclusions of the manuscript describe the advantages of the different types of cover crops separately, but still do not clearly indicate which treatment is the best to use to guide production practices based on a combination of indicators. It is recommended to add.

Reviewer 4 Report

This paper has a good concept with sufficient and advanced analysis and modeling. However, the highly varied data and the inconsistency of the experiment design makes the results highly unreliable. Modeling is trying to explain a general conclusion, which I found myself hard to learn the general and novel conclusions from your data and analysis.

Table 3, 4: combining two-year data is an easy way to show your results, but it’s really skeptical especially you have a very different two-year environment and even different crops and different time schedule. You have a very high variation in your results, I suggest you separate the results by years, not combined.

Line 298-303: why do you remove cover crop treatment and year effects from your full mixed model, are they not significant? Please show your full model. When you have an extremally different environment (precipitation) and different crops with potential different nitrogen use patterns, how can you combine the two years data and say “their responses to the cover crop treatments were comparatively similar”. Can you show examples and explain this?

Figure 2: by seeing this figure, I realized your data are very separated, especially for the 1st year, I highly recommend you have a third-year evaluation to reduce the variation in order to have a more common conclusion instead of the random conclusion.

Figure 3: PCA plot, PC1 and PC2 combined can explain less than 50% variation, that’s because you have a varied data, again, a third-year evaluation is recommended.

Reviewer 5 Report

The structure design of this paper is very reasonable, but there is a serious shortcoming, that is, the inter-annual repetition is too poor. The results and conclusions are seriously insufficient. If the author can't revise or explain well, I can only recommend rejecting the manuscript, although it is a pity. See below for some other specific comments.

Line 18-20. Why?

Line 20. Is monoculture inappropriate? Planting Forage oat while fallow?

Line 44. The definition of cover crops does not seem to be entirely accurate, and cover crops can also have the same growth period with the main crops.

Line 57-70. In water-deficient areas, there should be few farmers who choose to plant cover crops to achieve the purpose of water conservation, and most farmers choose to use film or straw mulching to retain water.

Line 63. Maybe have extra spaces.

Line 76-80, why different results arised in your study?

Figure 1. What does CC mean? Also, I don't think it's a good presentation. Very messy, however, the important information is not clear. It may be clearer to display it in the form of a time axis or an area chart. Please to delete unnecessary information, for example, 1990-2019.

Table 1. I think it's easy to be confused by your % here. Do the numbers in the table have percentages? But in fact it should be the amount of seeds usage, right?

Table 2. The row numbers in the table I think are redundant, no fertilization at all for the second year? normal operation? This fertilizer management scheme is not well understood.

Other tables and figures. Any abbreviations require a note for explanation.

Formula [1]. Are the units of the items in the formula the same? How are SWS and SWH obtained? What are the units? 

Table 3 and Table 4. Tables 3 and 4 should belong to the Results and should not be placed in the Materials and Methods. What is the significance of such results without statistical analysis?

Results section. The two-year results are so different that I don't recommend it for publication. The result is trouble for yourself and trouble for readers. The results of the PCA analysis are even more revealing of how huge the difference between the two-year results is. For example, SNAC and GRMN have a negative correlation in the first year and a positive correlation in the second year. For example, in the first year Fallow and Stubble were significant outliers, while in the second year, only Fallow was an outlier. In addition, based on the above reasons, how reliable can the SEM results be? It's hard to judge and hard to convince.

Discussion. You also don't analyze inter-annual differences in your discussion, trying to convince readers of your conclusions. This surprised me and deepened my concerns about this darft. Luckily, you have a lot of trial treatments, so I recommend only using the results from the first year, which also happens to be dry and water-deficient.

Round 2

Reviewer 3 Report

Thank you to the authors for their thorough replies to address my concerns. The revised manuscript has been significantly enhanced.

Reviewer 4 Report

Just want to response to authors’ responses as they mentioning “While we agree a third year of evaluation would be valuable, we are not in a position to do this. Project funding only lasts for so long!”

I highly doubt if the author could understand the valuable about a multiyear field experiment, I didn’t see any improvement from the manuscript, if you just want to publish whatever you have, even no any reliable and useful information, then please tell editor to do so. Publication is a very serious thing, you don’t have funding, then you need to resolve that problem, if you think you can publish any paper with some sort of data analysis, then you are wrong from my perspective. I have to reject this manuscript because I don’t want this kind of publication to mislead readers!

Reviewer 5 Report

I think the author has his own attitude, but this atttitude is not scientific. It can be seen from the author's response that the authors still lacks the exercise of scientific writing, including the understanding of scientific issues, as well as the scientific interpretation, scientific expression of the results. The author enven dont't understand that multi-year (>1 years) expriments need to be welll replicated. Since some of the suggestions I gave were rejected by the author, I don't think this version can be published. Because if such an article were to be published, it would mislead readers. Good luck.